# Suicidal ideation, attempt and associated factors among prisoners in Northwest Ethiopia: A cross-sectional study

Setegn Fentahun [ORCID]*, Mesele Wondie, Mamaru Melkam, Gebresilassie Tadesse, Getachew Tesfaw

Department of Psychiatry, College of Medicine and Health Science, University of Gondar, Gondar, Ethiopia

* Setegnf21@gmail.com

## Abstract

### Background

Suicide is a prominent source of harm and death globally, and it is the leading cause of premature death among prisoners. Therefore, the main aim of this study was to determine the prevalence and factors associated with suicidal ideation and attempt among prisoners in Northwest Ethiopia.

### Methods

An institution-based cross-sectional study design was performed from May 23 to June 22, 2022. After proportional allocation to the three correctional institutions, a total of 788 study participants were randomly recruited. The World Health Organization Composite International Diagnostic Interview (CIDI) was used to evaluate suicide ideation and attempt. To determine factors associated with suicidal ideation and attempt, multivariate logistic regression analyses were conducted. At a 95% confidence interval (CI) of $P$-value <0.05, statistical significance was declared.

### Results

The prevalence of suicidal ideation and attempt among prisoners was 23.6% and 10.7%, with 95% CI (20.76, 26.70) and (8.68, 13.02), respectively. Female sex (AOR = 2.38, 95% CI: 1.12, 5.05), family history of mental illness (AOR = 3.09, 95% CI: 1.93, 4.88), depression (AOR = 2.04, 95% CI: 1.43, 2.98), poor social support (AOR = 2.76, 95% CI: 1.56, 4.85) and previous incarceration (AOR = 1.84, 95% CI: 1.18, 2.86) were significantly associated with suicidal ideation. However, being single (AOR = 2.58, 95% CI: 1.47, 4.54), family history of suicide (AOR = 2.43, 95% CI: 1.18, 5.01), depression (AOR = 2.62, 95% CI: 1.59, 4.31) and previous imprisonments (AOR = 2.11, 95% CI: 1.20, 3.69) were associated with suicidal attempt.

**Data Availability Statement:** All relevant data are within the paper and its Supporting Information files.

**Funding:** The author(s) received no specific funding for this work.

**Competing interests:** The authors have declared that no competing interests exis.

**Abbreviations:** AOR, Adjusted Odds Ratio; CI, Confidence Interval; CIDI, Composite International Diagnostic Interview; COR, Crude Odds Ratio; DSM, Diagnostic and Satirical Manual; PHQ, Patient Health Questionnaires; USA, United States of America; WHO, World Health Organization.

## Conclusions

The prevalence of suicidal ideation and attempt among prisoners were found to be high. Therefore, the findings of this study recommend that early detection and design of prison mental health services should be delivered to develop mental health care, prevention, and intervention programs for incarcerated people to improve suicidal behavior in prison.

## Introduction

Suicide is described as the intentional termination of one's own life, and it is one of the main causes of death and significant public health issues around the world [1, 2]. There is a difference between thinking about suicide and doing it; some people have suicidal thoughts but never act on it [3]. Suicidal ideation, defined as having thoughts, desires, or wishes to terminate one's own life, is the beginning of the suicidal process. While a suicide attempt is a self-destructive act committed with the goal of ending one's own life [4, 5].

Suicidal behavior is a serious global public health concern that has an extremely negative influence on society's most vulnerable individuals, especially those who are living in prison [6, 7]. Over 800,000 people worldwide commit suicide each year, and it is the second leading cause of death for those between the ages of 15 to 29 [8]. More than ten million people are incarcerated throughout the world, and more than two-thirds of the world's countries have seen an increase in the number of imprisoned people [9]. In the United States, incarcerated people are nearly four times more likely to commit suicide than the general population, and nearly half of the prisoners have had suicidal ideation or conduct at some point in their lives, and one in every five has attempted suicide [10].

Based on the 2021 World Prison Population List Report, 110,000 individuals are imprisoned in Ethiopia [11]. The suicide rate in prison is higher than that of people of a similar age and sex who live in the community, with a nine-fold increase in the risk of suicide and a two-fold increase in the actual suicide rate [9]. The majority of mortality studies have focused on deaths that occur while in custody and have found that prisoners are more likely to die prematurely and are a more vulnerable group than people who have never been in prison. This makes suicide the leading and most common cause of death among prisoners [7, 9, 12, 13].

According to international research comparing suicide rates in prisons, the Nordic countries have the highest suicide rates in the world [14]. A meta-analysis study of 27 countries showed that 41.4% of deaths in prison are associated with suicide [13, 15]. According to a recent meta-analysis of prison suicides in 24 counties, Norway was found to have a substantially higher risk of suicide among prisoners than the general population, with 180 suicides per 100,000 prisoners [14].Suicide accounts for almost 40% of all deaths in South African prisons, making it the most common cause of unnatural death among inmates [16].

The mental health morbidity rate among prisoners is extremely high [17]. Individuals with mental illnesses are at an increased risk of suicide in high-income countries; people with mental illnesses account for over 90% of all suicides [7, 18]. The psychological effects of imprisonment have a major impact on prisoners' psychological well-being, As a result, prisoners are more likely to consider suicide [19]. Suicidal behavior has negative emotional, physical, and financial impacts. Prisoners who attempt suicide and survive may experience serious injuries that can have long-term effects on their health. It also affects the health and well-being of friends, loved ones, families, and the community [17].

Different factors contribute to suicidal ideation and attempts among prisoners. Attempting suicide before incarceration, genetics, perceived stigma, and certain personality traits such as impulsiveness are all risk factors for prisoner suicide [10, 20, 21]. It is also due to the types of crimes committed before incarceration and environmental factors, such as the type and quality of prison environments [22]. Different periods of imprisonment and release have been linked to an increased risk of suicide, and the first weeks of incarceration are linked to increased risks, particularly among young males [23, 24].

Studies have shown that the majority of inmates worldwide reside in low- and middle-income countries, including Ethiopia [25]. The Ethiopian prison system is structured in one federal prison system and ten regional prison systems. There are several prisons in Ethiopia, and a large number of prisoners are incarcerated; most of them are in the productive age group. Prison in Ethiopia remains harsh and, in some cases life threatening. The most common problems in Ethiopian prisons are extreme overcrowding, insufficient food and water, and poor healthcare. Numerous inmates had severe health problems; however, they got little or no treatment. Since Ethiopia lacks a well-integrated national health system and prison health system, inadequate mental health services are offered in prisons, and few prisoners have the opportunity to receive psychiatric care.

Despite suicide being the most serious problem among prisoners, which is also likely becoming more common as it is in other developed nations, it is an underreported health problem and has received little attention in developing countries, especially in Ethiopia. Moreover, the magnitude of suicidal ideation and attempts and the factors that contribute to them have not been extensively studied. Therefore, this study aimed to assess the prevalence and associated factors of suicidal ideation and attempts among prisoners in Northwest Ethiopia.

## Methods and materials

### Study area and population

An institution-based cross-sectional study design was conducted from May 23 to June 24, 2022. The study was conducted in the South Gondar Zone Correctional Centers. It is located in the Amhara regional state, 666 kilometers from Addis Ababa, Ethiopia, and 99 km from Bahir Dar (the capital city of the Amhara region). According to the Ethiopian Central Statistical Agency's 2007 Census, this zone had a total population of 2,051,738, with 1,041,061 men and 1,010,677 women. The zone consists of 11 districts and has seven primary and one referral hospitals. There are three correctional centers in this zone namely, Addis Zemen, Debre Tabor, and Gaynit town correctional centers. In each prison, there is a professional nurse who gives first aid to prisoners, but there is no psychiatry professional (psychiatric service) in the three correctional centers. Prisoners get health care services from nearby hospitals and the prison covers the cost of health care. According to data obtained from the prisons' head office, the three correctional centers had a total of 2513 prisoners during the study period. All prisoners in prison aged 18 years and older during the data collection period were included in this study. Whereas, Prisoners who are critically sick and unable to communicate (unable to speak and unable to hear) during data collection were excluded from the current study.

### Sample size determination and sampling procedure

The sample size was determined by assuming a single proportion formula. The prevalence of suicidal ideation and attempt were 21.9% and 13.1% in the previous study conducted in Ethiopia, respectively [26], with a 95% confidence interval (CI), margin of error 3%, and 10% non-response rate. 803 prisoners were recruited randomly using stratified random sampling to allocate a sample of prisoners in each correctional center During the study period, there were

2513 total prisoners (813, 1100, and 600 in Adiss Zemen Correctional Center, Debre Tabor Correctional Center, and Gaynit correctional center, respectively). Accordingly, for each correctional center, a proportional allocation of the study participants was calculated. As a result, 260 participants from Adiss Zemen, 351 from Debre Tabor, and 192 study participants from Gaynit correctional center were selected. Lastly, a simple random sampling technique was used to select participants from each correctional center by a computer-generated method of their identification number.

$n = ((Z\alpha/2)^2 p (1-p))/d^2$ $n = ((1.96)2 \times 0.219 \times .781))/(0.03)^2 = 730$

Including 10% of the non-response rate, the final sample size was 803.

## Data collection tools

Data were collected using an interviewer-administered questionnaire, which contained socio-demographic characteristics, clinical-related factors, ever-use substance, social support, perceived stigma, prison-related factors, and suicidal ideation and attempt.

Social support was assessed by Oslo 3-item social support scale with ranging scores from 3–14 and had three broad categories: "poor support" 3–8, "moderate support" 9–11, and "strong support" 12–14 [27]. Perceived stigma was measured by Jacoby's 3-item perceived stigma scale with a score of one and above considered as having perceived stigma [28].

Depression among prisoners was assessed by using a Patient Health Questionnaire (PHQ-9). It is a nine-item version, each item response is rated as "0" (not at all) to "3" (nearly every day), and the total score ranges from 0 to 27 with a cut-off $\geq 5$ have depression symptoms. It has a sensitivity of 88% and a specificity of 88%. PHQ-9 has been translated and validated in Ethiopia and it has been extensively used in Ethiopia previously to assess depression [29]. Anti-social personality disorder was measured using DSM−5 section II ASPD Diagnostic criteria symptoms. The ASPD criteria symptoms have seven items, is an assessment of ASPD accordingly DSM-V which asks participants to respond either 'Yes or 'No' to every 7 items, scored three points or above from the seven questions is considered as having an ASPD and cut point less than three not having an ASPD [30].

Suicidal ideation and attempt were measured according to the World Mental Health (WMH) survey initiative version of the World Health Organization (WHO) Composite International Diagnostic Interview (CIDI). It is a standard tool used in different studies in Ethiopia to assess suicide ideation and attempts in different population groups and settings including both clinical and community levels [26, 31]. Based on this tool, if the respondents answer "Yes" to the question "Have you seriously thought about committing suicide after you are imprisoned?" were considered to have had suicidal ideation, and the respondents who answer "Yes" to the question "Have you attempted committing suicide after you are imprisoned?" were considered as having suicide attempt [32].

## Data quality control

To control the quality of the data, the questionnaire was initially prepared in English, translated into the local Amharic language, and then back-translated to English by two language experts to maintain consistency. The questionnaire was pre-tested one week prior to the actual data collection time on 5% (n = 40) of the study who were not included in the main study and collected by six BSc nurse professionals and two BSc psychiatry profession supervisors using interviewer-administered questionnaires. For those data collectors and the supervisors, one day of training was given before the actual data collection. The internal consistency of the suicidal ideation and attempt tool (CIDI) was checked using Cronbach's alpha (α = 0.91), which

shows an acceptable level of internal consistency. Based on the pre-test result, an appropriate modification was made to the questionnaire.

**Data processing and analysis.** Data were coded and entered into the computer using EPI data version 4.6.02 and exported to STATA version 14 for analysis. Descriptive statistics were presented using percentages and frequencies. To determine an association between dependent and independent variables adjusted odds ratio was used using logistic regression and the significance level was determined using a confidence interval of 95%. Bivariate and multivariate analysis was done to identify the independent predictors of outcome variables. This was done by entering each independent variable separately into the bivariate analysis. Those variables with a p-value of less than 0.2 on bivariate analysis were entered into multivariate analysis for further analysis. Then, an adjusted odds ratio with 95% CI was computed for variables having a p-value less than 0.05 in multivariate analysis and was considered as predictors of outcome variables.

**Ethics statement.** Ethical approval was obtained from the ethical review board of the University of Gondar. A formal letter of permission was obtained from the Department of Psychiatry. Then it was submitted to the head office of South Gondar correctional institutions and each prison institution to get permission to collect data on the correctional site. Participants were informed that the study must be confidential and have no negative impact on their lives. After explaining the objectives of the study, written informed consent was obtained from the participants prior to data collection. The interview was given only to consenting participants. They were told that they had been randomly selected to participate in the study and that they could leave the interview at any time. Participants were informed that their choice to participate in the study or refuse it would not affect the duration of their imprisonment or their possibility of release. To ensure privacy, prisoners were interviewed away from others during the interview. Throughout the study, confidentiality and anonymity were guaranteed for all personal information. The collected data was confidentially secured and stored.

## Results

### Socio-demographic characteristics of the study Participants

A total of 788 study participants were involved in the current study with a response rate of 98.2%. The majority 752 (95.4% of the respondents were male. The mean age of the study participants was 34.3 (±13.9) years. Out of the respondents, 322 (40.9%) were in the age group of 18–27 years; almost half (51.6%) of them were married; most of the respondents (98.5%) were Orthodox Christians; and about one-third of the 249 (31.6%) participants were unable to read and write. Nearly four-fifths (80.8%) of respondents were from rural areas; more than half (57.4%) of prisoners were farmers; and 613 (77.8%) had no work inside the prison (Table 1).

### Clinical related factors

Among the participants, 320 (40.6%) and 128 (16.2%) of the prisoners had comorbid depression and antisocial personality disorder, respectively. Out of the study participants, 95 (12.1%) had a history of mental illness, 105 (13.3%), and 65 (8.3%) respondents had a family history of mental illness and suicide attempt, respectively. Nearly one-fourth, 202 (25.6%) of study participants had a history of ever using a substance, and 97 (12.3%) had a history of chronic physical illness (Table 2).

### Criminal and psychosocial characteristics of study participants

Out of the total participants, 536 (68.1%) got the decision of the courts, and the rest of the 252 (31.9%) have been imprisoned without getting the court's decision. Out of the respondents,

**Table 1. Socio-demographic characteristic distribution among prisoners in Northwest Ethiopia, 2022 (n = 788).**

| Variables | Categories | Frequency | Percentage |
|---|---|---|---|
| | Male | 752 | 95.4 |
| Sex | Female | 36 | 4.6 |
| Age | 18–27 | 322 | 40.9 |
| | 28–37 | 206 | 26.1 |
| | 38–47 | 117 | 14.9 |
| | ≥48 | 143 | 18.1 |
| Marital status | Single | 272 | 34.5 |
| | Married | 407 | 51.6 |
| | Divorced | 84 | 10.7 |
| | Widow/widower | 25 | 3.2 |
| Religion | Orthodox | 776 | 98.5 |
| | Protestant | 5 | 0.6 |
| | Muslim | 7 | 0.9 |
| Educational Level | Unable to read and write | 249 | 31.6 |
| | Able to read and write | 214 | 27.2 |
| | Educated up to grade 8th | 105 | 13.3 |
| | Grade 9-12th | 140 | 17.7 |
| | Diploma | 37 | 4.7 |
| | Degree and above | 43 | 5.5 |
| Residency | Urban | 151 | 19.2 |
| | Rural | 637 | 80.8 |
| Previous occupation | Gov.t employed | 34 | 4.3 |
| | Unemployed | 59 | 7.5 |
| | Self-employed | 28 | 3.6 |
| | Farmer | 452 | 57.4 |
| | Student | 115 | 14.6 |
| | Merchant | 63 | 7.9 |
| | Others* | 37 | 4.7 |
| Work inside prison | Yes | 175 | 22.2 |
| | No | 613 | 77.8 |

*Daily laborer, driver and Garage worker

245 (45.7%) were sentenced to more than ten years, and nearly two-thirds, 522 (66.2%) of respondents were imprisoned for less than two years. The majority of participants, 424 (53.8%), had been charged because of murder, followed by 194 (24.6%) with theft and robbery. Of the total participants, 137 (17.4%) had a history of imprisonment, and the major frequency of previous incarceration, 91(66.4%) was once. Nearly half (412 (52.3%) of respondents had perceived stigma, whereas 376 (47.7%) had no perceived stigma. Among the total participants, 410 (52%), 242 (30.7%), and 136 (17.3%) had poor, moderate, and strong social support, respectively (Table 3).

## The prevalence of suicide ideation and attempt

The prevalence of suicide ideation and attempt among prisoners after imprisonment was 186 (23.6%) and 84 (10.7%), with a 95% CI (20.76, 26.70) and (8.68, 13.02), respectively. The prevalence of suicidal ideation and attempt in the last month was 65 (8.3%) and 27 (3.4%), respectively. Of the participants, 121 (15.4%) had planned to commit suicide during the entire period

**Table 2. Clinical and substance characteristics among prisoners in Northwest Ethiopia, 2022 (n = 788).**

| Variables | Categories | Frequency | Percentage |
|---|---|---|---|
| History of mental illness | Yes | 95 | 12.1 |
| | No | 693 | 87.9 |
| Family history of mental illness | Yes | 105 | 13.3 |
| | No | 683 | 86.7 |
| Family history of suicide attempt | Yes | 65 | 8.3 |
| | No | 723 | 91.7 |
| Comorbid depression | Yes | 320 | 40.6 |
| | No | 468 | 59.4 |
| Antisocial personality disorder | Yes | 128 | 16.2 |
| | No | 660 | 83.8 |
| Chronic physical illness | Yes | 97 | 12.3 |
| | No | 691 | 87.7 |
| Ever use of substance | Yes | 202 | 25.6 |
| | No | 586 | 74.4 |

**Table 3. Criminal and psychosocial characteristics among prisoners in Northwest Ethiopia, 2022 (n = 788).**

| Variables | Categories | Frequency | Percentage |
|---|---|---|---|
| Court's decision | Yes | 536 | 68.1 |
| | No | 252 | 31.9 |
| Total sentence (in years) | <2 | 68 | 12.7 |
| | 2–5 | 76 | 14.2 |
| | 6–10 | 147 | 27.4 |
| | >10 | 245 | 45.7 |
| Duration in the prison (in years | <2 | 522 | 66.2 |
| | 2–5 | 195 | 24.8 |
| | 6–10 | 64 | 8.1 |
| | >10 | 7 | 0.9 |
| | Murder | 424 | 53.8 |
| | Physical attack & try to murder | 102 | 12.9 |
| Types of crime | Rape & abduction | 41 | 5.2 |
| | Theft & Robbery | 194 | 24.6 |
| | Corruption | 8 | 1.1 |
| | Others* | 19 | 2.4 |
| Previous incarceration | Yes | 137 | 17.4 |
| | No | 651 | 82.6 |
| Number of previous incarceration | Once | 91 | 66.4 |
| | Twice | 43 | 31.4 |
| | Three & above | 3 | 2.2 |
| Perceived stigma | Yes | 412 | 52.3 |
| | No | 376 | 47.7 |
| Social support | Poor | 410 | 52.0 |
| | Moderate | 242 | 30.7 |
| | Strong | 136 | 17.3 |

* deception, surety, and kidnap

Table 4. Prevalence of suicidal ideations and attempts among prisoners in Northwest Ethiopia, 2022 (n = 788).

| Variables | Categories | Frequency | Percentage |
|---|---|---|---|
| Ever suicidal ideation | Yes | 186 | 23.6 |
| | No | 602 | 76.4 |
| Suicidal ideation in the last one month | Yes | 65 | 8.3 |
| | No | 723 | 91.7 |
| Ever plan of suicide | Yes | 121 | 15.4 |
| | No | 667 | 84.6 |
| Ever suicide attempt | Yes | 84 | 10.7 |
| | No | 704 | 89.3 |
| Suicide attempt in 1 month | Yes | 27 | 3.4 |
| | No | 761 | 96.6 |
| Frequency of suicide attempt | Once | 59 | 70.2 |
| | Twice | 20 | 23.8 |
| | Three and more | 5 | 6.0 |
| | Serious attempt | 56 | 66.7 |
| Severity related to attempt | Not foolproof | 24 | 28.5 |
| | Not intend to die | 4 | 4.8 |
| | Family conflict | 11 | 13.0 |
| Reason for suicide attempt | Feel guilty of crime committed | 15 | 17.9 |
| | Death in family | 17 | 20.2 |
| | Financial loss/poverty | 10 | 11.9 |
| | Mental illness | 3 | 3.6 |
| | Physical illness | 2 | 2.4 |
| | Hopelessness due to crime | 26 | 31.0 |

of their imprisonment. Out of those who attempt suicide, the majority, 59 (72.2%) of respondents, attempt suicide once. The most frequent method used for suicide attempts was hanging 31 (36.9%). The major reasons for suicide attempts were hopelessness due to crime 26 (31%) followed by the deaths of family members 17 (20.2%) (Table 4).

## Factors associated with suicidal ideation

As revealed from the bivariate logistic regression analysis, sex, marital status, occupation, history of mental illness, family history of mental illness, comorbid depression, perceived stigma, social support, and previous incarceration were significantly associated with suicidal ideation at a p-value < 0.2. In the multivariate logistic regression analysis, female sex, family history of mental illness, co-morbid depression, poor social support, and previous incarceration were significantly associated with suicidal ideation at p-value < 0.05. The female sex was 2.4 times more likely to have suicide ideation compared with the male sex (AOR = 2.38, 95% CI: 1.12, 5.05). Those participants who had a family history of mental illness was 3.1 times greater odds of having suicidal ideation compared to prisoners without a family history of mental illness (AOR = 3.09, 95% CI:1.93, 4.88). Those who had co-morbid depression were 2 times more likely to have suicidal ideation compared to those who had no co-morbid depressive symptoms (AOR = 2.04, 95% CI: 1.43, 2.98). Poor social support was about 2.8 times more likely to develop suicide ideation than participants who had strong social support (AOR = 2.76, 95% CI: 1.56, 4.85). Furthermore, participants who had previous incarceration were 1.8 times more likely to have suicidal ideation than prisoners incarcerated in prison with no previous history of prison (AOR = 1.84, 95% CI: 1.18, 2.86) (Table 5).

**Table 5. Bivariate and multivariate logistic regression analysis of suicidal ideation and associated factors among prisoners in Northwest Ethiopia, 2022 (n = 788).**

| Variables | Categories | Suicidal ideation | | COR, (95% CI) | AOR, (95% CI) |
|---|---|---|---|---|---|
| | | Yes | No | | |
| Sex | Male | 171 | 581 | 1.00 | 1.00 |
| | Female | 15 | 21 | 2.43 (1.22, 4.81) | 2.38(1.12, 5.05)* |
| Marital status | Married | 82 | 325 | 1.00 | 1.00 |
| | Single | 70 | 202 | 1.37(0.95, 1.98) | 1.33(0.86 2.04,) |
| | Divorced | 28 | 56 | 1.98(1.18, 3.31) | 1.68(0.94, 2.97) |
| | Widow/widower | 6 | 19 | 1.25(0.48, 3.23) | 0.89(0.32, 2.55) |
| Occupation | Employed | 6 | 28 | 1.00 | 1.00 |
| | Unemployed | 23 | 36 | 2.98(1.07, 8.31) | 2.95(0.96, 9.04) |
| | Self-employed | 9 | 19 | 2.21(0.68, 7.23) | 2.46(0.66, 9.12) |
| | Farmer | 89 | 363 | 1.14(0.46, 2.85) | 1.26(0.47, 3.38) |
| | Student | 29 | 86 | 1.57(0.59, 4.18) | 1.74(0.59, 5.11) |
| | Merchant | 21 | 42 | 2.33(0.84, 6.51) | 2.08(0.67, 6.39) |
| | Others | 9 | 28 | 1.52(0.47, 4.78) | 1.17(0.32, 4.18) |
| History of mental illness | Yes | 64 | 31 | 1.68(1.06, 2.68) | 1.25(0.74, 2.11) |
| | No | 538 | 155 | 1.00 | 1.00 |
| Family history of mental illness | Yes | 52 | 52 | 4.12(2.67, 6.29) | 3.07(1.93, 4.88)*** |
| | No | 134 | 550 | 1.00 | 1.00 |
| Comorbid Depression | Yes | 86 | 382 | 2.02(1.45, 2.82) | 2.06(1.43, 2.98)*** |
| | No | 100 | 220 | 1.00 | 1.00 |
| Perceived stigma | Yes | 109 | 303 | 1.40(1.00, 1.95) | 1.27(0.88, 1.84) |
| | No | 77 | 299 | 1.00 | 1.00 |
| Social support | Poor | 134 | 276 | 2.98(1.77, 5.06) | 2.76(1.56, 4.85)*** |
| | Moderate | 33 | 209 | 0.97(0.53, 1.78) | 0.87(0.46, 1.64) |
| | Strong | 19 | 117 | 1.00 | 1.00 |
| Previous incarceration | Yes | 47 | 90 | 1.92(1.29, 2.87) | 1.84(1.19, 2.89)** |
| | No | 139 | 512 | 1.00 | 1.00 |

*P Value < 0.05

**P value < 0.01, and

***P value < 0.001, 1 = reference, Hosmer and Lemeshow test = 0.85

## Factors associated with suicidal attempt

The bivariate logistic regression analysis showed that marital status, previous occupation, comorbid depression, family history of mental illness, family history of suicide attempt, history of mental illness, and previous incarceration were significantly associated with suicidal ideation at p-value < 0.2. The result of multivariate logistic regression analysis revealed that prisoners who had a family history of suicidal attempt, depression, prior incarceration, and being single were significantly associated with suicidal attempt at p-value < 0.05. Being single was 2.6 times the risk of suicidal attempt compared to married participants (AOR = 2.58, 95% CI: 1.47, 4.54). Family history of suicidal attempt was 2.4 times greater risk of suicidal attempt compared to their counterparts (AOR = 2.43, 95% CI: 1.18, 5.01) and co-morbid depression also had 2.6 times higher risk of developing suicide attempt compared to prisoners who did not have depression (AOR = 2.62, 95% CI: 1.59, 4.31). Finally, the odds of developing suicide attempt were 2.1 times higher among prisoners who had previous incarcerations than participants without previous incarcerations (AOR = 2.11, 95% CI: 1.20, 3.69) (Table 6)

**Table 6. Bivariate and multivariate logistic regression analysis of suicide attempt and associated factors among prisoners in Northwest Ethiopia, 2022 (n = 788).**

| Variables | Categories | Suicidal attempt | | COR, (95% CI) | AOR, (95% CI) |
|---|---|---|---|---|---|
| | | Yes | No | | |
| | Married | 30 | 377 | 1.00 | 1.00 |
| Marital status | Single | 41 | 231 | 2.23(1.35, 3.67) | 2.60(1.48, 4.58)** |
| | Divorced | 11 | 73 | 1.89(0.91, 3.94) | 1.83(0.82, 4.07) |
| | Widow/widower | 2 | 23 | 1.09(0.25, 4.86) | 0.97(0.21, 4.65) |
| Previous occupation | Employed | 4 | 30 | 1.00 | 1.00 |
| | Unemployed | 19 | 40 | 3.56(1.09, 11.56) | 2.92(0.85, 10.01) |
| | Self-employed | 4 | 24 | 1.25(0.28, 5.53) | 1.13(0.24, 5.35) |
| | Farmer | 41 | 411 | 0.74(0.25, 2.23) | 0.92(0.29, 2.86) |
| | Student | 10 | 105 | 0.71(0.21 2.44,) | 0.57(0.56, 2.08) |
| | Merchant | 5 | 58 | 0.65(0.16, 2.58) | 0.52(0.12, 2.26) |
| | Others | 1 | 36 | 0.21(0.02, 1.97) | 0.17(0.02, 1.67) |
| Depression | Yes | 53 | 267 | 2.79(1.75, 4.47) | 2.62(1.61, 4.34)*** |
| | No | 31 | 437 | 1.00 | 1.00 |
| Family history of mental illness | Yes | 18 | 86 | 1.95(1.11, 3.46) | 1.79(0.97, 3.31) |
| | No | 66 | 618 | 1.00 | 1.00 |
| Family history of suicide attempt | Yes | 12 | 53 | 2.05(1.06, 4.01) | 2.43(1.18, 5.03)* |
| | No | 72 | 651 | 1.00 | 1.00 |
| History of mental illness | Yes | 16 | 79 | 1.86(1.03, 3.37) | 1.65(0.87, 3.17) |
| | No | 68 | 625 | 1.00 | 1.00 |
| Previous incarceration | Yes | 23 | 114 | 1.95(1.16, 3.28) | 2.15(1.22, 3.76)** |
| | No | 61 | 590 | 1.00 | 1.00 |

*P Value < 0.05

**P value < 0.01, and

***P value < 0.001, 1 = reference, Hosmer and Lemeshow test = 0.48

## Discussion

In this study, the prevalence of suicidal ideation and attempt among prisoners and potential correlations with different variables have been examined. The prevalence of suicidal ideation and attempt during their current imprisonment was 23.6% and 10.7% with 95% CI (20.76, 26.70) and (8.68, 13.02), respectively. Concerning the prevalence of suicidal ideation, the magnitude of suicidal ideation in the current study was in line with the other studies reported that were carried out among prisoners in Ethiopia 21.9% [26], United States of America (23.5%) [6] and Pakistan (22%) [33].

Contrarily, the prevalence of suicidal ideation in this study was higher than in the two studies that were previously conducted in Ethiopia: 8.04% in Addis Abba Correction Centre [34] and 16.6% in Jimma Correctional Institution [35] and other studies done in Cambodia (7%) [36], Chicago (10%) [21], USA (9.4%, 15.1%) [37, 38], New South Wales, Australia (16%) [39], Taiwan (12.5%) [40] and Colombia (14.9%) [41]. The possible reason for the observed difference might be assessment tool differences. For instance, Attitude Toward Suicide (ATTS), Diagnostic Interview Schedule for Children (DISC), Brief Symptom Rating Scale (BSRS-5), and Inventory of Suicide Orientation (ISO-30) were used in Cambodia, Chicago, Taiwan, and Colombia respectively. The additional discrepancy might be due to variation in study participants, socio economic status and cultural difference of participants. For example, in Cambodia and Taiwan, the studies were conducted only among male prisoners who had a lower

likelihood of developing suicidal ideation [13, 26]. Another possible reason may be related to the difference in health service delivery in correctional centers and the variation in prison environments [19, 22]. In Ethiopia, there are inadequate mental health services in correctional facilities, and the prison environments are extremely harsh, which might increase the prevalence of prisoners' suicidal ideation and attempts.

However, the current finding was lower than the study conducted in New Zealand 34.6% [42], Iran 38.2% [43], USA 38.7% [44], Belgium 32%, 44% [45, 46], and China 33% [47]. The discrepancy might be due to the difference in assessment tools and sampled study participants. For example, in China, only female prisoners were sampled who had a high chance of developing suicidal ideation [3] and the tool they used was the Beck Scale for Suicidal Ideation-Current (BSSI-C). Another possible reason for the observed variation might be the sociocultural variation of the respondents since suicidal behaviors are strongly influenced by sociocultural factors [7].

Regarding the magnitude of suicidal attempt, according to the findings of this study, the prevalence of suicidal attempt among prisoners was 10.7%. This is closely consistent with previous studies reported in Ethiopia 9.3% [35] and 13% [26], Australia 10% [39], Nigeria 8% [48], Belgium 9% [45], and Spain 8.7% [49].

However, this result was higher than the study conducted in different countries. In a study conducted in Cambodia among 572 young male prisoners aged between 15 and 24 using the Attitude Toward Suicide (ATTS), the prevalence of suicide attempts was 3%, which was lower than the result of the current study [36]. The finding of this study was higher than a study carried out among 535 HIV-infected male prisoners in Taiwan using the Brief Symptom Rating Scale (BSRS-5), with a 4.1% prevalence of suicide attempts [40].

Contrarily, the prevalence of suicidal attempt in this study was lower than in a systemic meta-analytic review conducted in South Wales, which was 23% [50]. It is also lower than other studies conducted in Iran 20.5% [43], USA 13.7%, 14%, and 23.3% [6, 37, 44], Italy male prisoners 23.7% [51], Belgium 22% [46], 25% in Spain male prisoners [52], 19.8% in Slovenia [53] and 18.8% in Brazil [54]. The possible reason for this difference might be the use of different screening tools. The study conducted in Slovenia was assessed using Paykel's Questions on Suicidal Behavior (PSS). However, the current study was done using the World Health Organization (WHO) Composite International Diagnostic Interview (CIDI). Another possible reason for the discrepancy could be the variation in study subjects. For instance, in Brazil, the study was conducted only among female prisoners. As different studies have revealed, females have been found to have a higher risk of suicide attempt than males [55].

Regarding associated factors of suicide ideation, this study found that women were considerably more likely than men to have suicidal ideation. The finding was supported by previous studies done in Ethiopia [26], Belgium [46], and Chicago [21]. This might be because women are more prone to suicidal behavior due to gender-related vulnerability to psychopathology and psychosocial stressors [56].

There was a statistically significant association between suicidal ideation and a family history of mental illness. This result was consistent with previous studies conducted in Ethiopia [26], Israel [57], and the USA [58]. This is due to a parental history of mental illness might increase the risk of suicidal ideation in the offspring through transmission of genetic factors [59]. In addition to this, suicidal behavior is more likely to occur when there is a hereditary predisposition and it can also be significantly controlled by biological and genetic factors [3].

Those participants who had comorbid depressive symptoms were significantly associated with suicidal ideation in this study. This finding was supported by previous studies conducted in Ethiopia [35], Taiwan [40], Israel [57], Italy [20], Chicago [21], and USA [37]. This could be due to the fact that depressed individuals have altered neurotransmitters that could contribute to worthlessness, guilt, and hopelessness and this may lead to suicidal behavior [3, 10].

Those respondents who had poor social support were predictors for suicidal ideation in this study. This is in line with previous studies reports that were carried out in Ethiopia [26, 35] and with other similar studies conducted in China [47], USA [13], and Belgium [60]. This could be due to having fewer close friends both inside and outside prison, feeling less close to relatives, and having fewer external contacts in the form of letters, phone calls, and visits are significant stressors for incarcerated individuals, and contribute to the prisoner's suicidal ideation [10]. According to a 2004 WHO report, suicide ideation has a strong correlation with weak social bonds and low support from friends or family [61]. Another possible reason that may contribute to suicidal ideation is a sense of isolation, low levels of perceived social support, and disruption in interpersonal relationships and support systems.

Prisoners who had previous incarceration were significantly associated with suicidal ideation. This finding was similar to reports of previous study in Ethiopia [35] and with other countries' studies conducted in Spain [56], Italy [20], and USA [37]. The possible reasons could be that repeated imprisonment may lead to withdrawal from social and family life; living alone, having fewer friends, low social ties, loneliness and exclusion from key community resources or activities may contribute to suicidal ideation in prisoners [10, 20].

Regarding factors associated with suicide attempt, those participants who were single are found to be more likely to have suicidal attempt when compared to married ones. This was supported by the study conducted in Colombia [41], England and Wales [62]. This might be due to the fact that marital relationships may affect mental health functioning in terms of financial well-being, and emotional and social support. Being single was found to be a more suicidal attempt because of a lack of social support and significant others to share emotional and other psychosocial burdens [13, 63].

This study showed that a family history of suicide was significantly associated with attempt suicide. This was in line with the studies done in Italy [51], China [47], and Belgium [46]. The possible reason might be that the risk of suicidal attempt is increased by genetic predisposition.

In addition to this, suicidal behavior is influenced by learned behavior, a common lifestyle, and shared exposure to familial stress and environmental influences [20, 64].

In this study, comorbid depression was found to be significantly associated with suicidal attempts, which is supported by studies conducted in different countries like Slovenia [53], Israel [57], Brazil [54], Italy [20], and Chicago [21]. This is due to the fact that prisoners are subjected to increased stress and hopelessness, which has a negative impact on their mental health and may exacerbate any psychopathology they already have, which has contributed to suicidal attempts [13, 42].

This research found that prisoners who had a history of incarceration were correlated with higher suicidal attempt than those who did not have incarceration before. This finding was supported by previous study carried out in Ethiopia [26] and other studies conducted in Italy [51] and the United States of America [44]. This may be related to the coping abilities of prisoners affected by the persistent stress of the prison environment, and the lack of proper mental health support in prison may contribute to suicide attempts [13].

## Limitations of the study

The cross-sectional nature of this study does not establish a real cause-and-effect relationship; it shows only a temporal relationship. Since the data was collected using an interviewer-administered questionnaire, this study was also susceptible to social desirability bias, specifically questions about criminal activity and substance abuse. In addition, it also leads to interviewer bias. The other limitation of the current study was that since the study was conducted

during their entire period of imprisonment and the source of data was based on retrospective self-report, it might have led to recall bias.

## Conclusion

The findings of the current study showed that the magnitude of suicidal ideation and attempt among prisoners was high. Female sex, family history of mental illness, depression, poor social support, and previous incarceration were significantly associated with suicidal ideation. Whereas, being single, having a family history of suicidal attempt, depression, and previous incarceration were significantly associated with suicidal attempt. Therefore, the findings of this study recommend that the Ethiopian Ministry of Health and the Federal Prison Administration should work together to strengthen the prison mental health program by expanding access to mental health services in correctional institutions. Early detection of suicidal ideation and attempts is crucial to reducing the impact and overall burden of suicidal ideation and attempts among incarcerated people. The results of the current study would be crucial for policymakers and healthcare interventions to improve prisoners' suicidal ideation and attempt. Finally, for future researchers, it is better to conduct interventional studies to demonstrate how to use interventional strategies to reduce the prevalence and risk factors of suicidal ideation and attempt in prisoners.

## Supporting information

**S1 Data set.**
(DTA)

## Acknowledgments

We would like to express our great gratitude to the South Gondar Prison administration for permitting the study and for their willingness to share vital information about the study area and population. Finally, Our deepest gratitude to the study participants who gave their precious time to participate voluntarily in this study and to data collectors and supervisors.

## Author Contributions

**Conceptualization:** Getachew Tesfaw.

**Formal analysis:** Setegn Fentahun.

**Methodology:** Setegn Fentahun, Mesele Wondie, Gebresilassie Tadesse.

**Supervision:** Setegn Fentahun.

**Writing – original draft:** Setegn Fentahun, Mamaru Melkam.

**Writing – review & editing:** Mesele Wondie.

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
