## [Decision Letter · Decision Letter 0]

19 Feb 2024

PONE-D-23-21802Suicidal ideation, attempt and associated factors among prisoners in Northwest Ethiopia: a cross-sectional studyPLOS ONE

Dear Dr. Fentahun,

Thank you for submitting your manuscript to PLOS ONE. After careful consideration, we feel that it has merit but does not fully meet PLOS ONE’s publication criteria as it currently stands. Therefore, we invite you to submit a revised version of the manuscript that addresses the points raised during the review process.

We look forward to receiving your revised manuscript.

Kind regards,

Vincenzo De Luca

Academic Editor

PLOS ONE

Journal Requirements:

2. Please provide additional information regarding the considerations  made for the prisoners included in this study. For instance, please discuss whether participants were able to opt out of the study and whether individuals who did not participate receive the same treatment offered to participants.

3. In the online submission form, you indicated that "All data are included in the manuscript. Data are available upon request from the corresponding author"

Reviewers' comments:

Reviewer's Responses to Questions

**Comments to the Author**

1. Is the manuscript technically sound, and do the data support the conclusions?

Reviewer #1: Yes

Reviewer #2: Yes

2. Has the statistical analysis been performed appropriately and rigorously? 

Reviewer #1: No

Reviewer #2: Yes

3. Have the authors made all data underlying the findings in their manuscript fully available?

Reviewer #1: Yes

Reviewer #2: Yes

4. Is the manuscript presented in an intelligible fashion and written in standard English?

Reviewer #1: Yes

Reviewer #2: Yes

5. Review Comments to the Author

Reviewer #1: Manuscript: PONE-D-23-21802

Title: “Suicidal ideation, attempt and associated factors among prisoners in Northwest Ethiopia: a cross-sectional study”

The paper evaluates the prevalence and associated factors of suicidal ideation and attempt among prisoners in Northwest Ethiopia in 2022 in order to inform health policies. The main statistical analysis is based on logistic regression.

This work is well-motivated and interesting for health research. As my expertise is limited to the methodology, there is not much to say as the statistical methodology is quite simple. The references needs to be double checked and some needs to be fixed. My comments and suggestions for possible modifications are below.

Comments

1. Abstract - Methods: The authors state "a simple random sampling technique was employed", this sounds too vague.

2. Abstract - Methods: "Binary and multivariate analyses were used". The authors could be a bit more specific here.

3. Introduction: The reference for "Over 800,000 people worldwide kill themselves each year (8)" is not correct, I believe it should be reference number 9.

4. Introduction: The reference for "Suicide rates in prison are seven times higher than in the general population (8)" does not seem to contain this information.

5. Introduction: Same comment for "In the United States, jailed people are roughly four times more likely to commit suicide than the general population, and nearly half of the prisoners have had suicidal ideation or conduct at some point in their lives, and one in every five has attempted suicide (10)."

6. Introduction: "Suicide rates in prison are seven times higher than in the general population"

and

"The suicide rate in prison is higher than that of people of a similar age and sex who live in the community, with a [...] two-fold increase in the actual suicide rate (11)." It seems inconsistent, this should be discussed...

7. Introduction: References 19 and 20 are identical to reference 15.

8. Sample size determination: "Including 10% of the non-response rate the final sample size was 730+73=803".

If you want to have 730 responses with an expected 10% of non-response rate, then you should do 730/0.9=811 instead of multiplying by 1.10 as 803 - 10% = 723...

9. Discussions: "This is due to a parental history of mental illness might increase the risk of suicidal ideation in the offspring through transmission of genetic factors (8)". Check the reference.

10. Conclusion: "the magnitude of suicidal ideation and attempt among prisoners was significantly high." I'm not sure of the meaning of "significantly high" here, maybe the authors should reformulate.

11. It is not clear if STATA was used for the logistic regression or just for descriptive statistics, this should be clarified.

12. Possible interactions between factors in multivariate logistic models are not discussed and could be of interest.

Reviewer #2: This research investigates various variables associated with ideations and attempts at suicide among prisoners in three prisons in North-West Ethiopia and the prevalence of ideations and attempts at suicide among the same prisoners. This research is in the main well conducted but several things should be better explained and the English in the text could be improved at places in the text. The weakest part of the text is, I think, the Discussion section which should be improved.

Line 91 (= L 91): delete the words “in the study areas”? Redundant!

It would be good if the authors could provide more info about the prison system I Ethiopia and NW Ethiopia and how the standard and routines of these compare to prisons in other more or less developed countries.

Moreover, I also missed a discussion in Discussion section how the standard of the prisons might have contributed to the suicide ideations and attempts. Please provide such a discussion!

L 107-108: Please provide info about How many prisoners there were there in total? And what it means that they were “included in the study”?

L 114-115: incomplete sentence!

L 115-122: This needs to be explained much better!

L 175: Why not “Results”?

L 177: Explain carefully what is meant by “response rate” here! For example, response rate in ration to what?

L 193: How can 45.7 % be “a majority”?

L 204: Since the prisoners were asked about suicide ideation and attempts only in the “last month” it seems likely that the rates of these measures will be under-reported if they are interpreted to be valid for a whole year. I assume the questions were asked like this for memory reasons, but the drawbacks and consequences should be discussed in the Discussion and under “limitations”!

Moreover, on L 243-244, given that they were asked about suicide ideation and attempts only in the “last month”, it does not seem appropriate to write about the “prevalence of suicidal ideation and attempt during their current imprisonment”.

Similarly, on L 261-262 it is not appropriate to write about “the entire period after their imprisonment”.

L 214 and L 216: Explain why different significance levels are used in these contexts!

L 241: should be “Discussion”!

L262: add the word “be”, or similar, in this sentence.

L 262-264: Poor grammar!

L 284-286: Too vague! Explain better how the difference could be due to “different instruments”! and “variation in study subjects”!

In the Discussion the results of the current study are related to previous research and then it is noted that they are similar to results from diverse studies in other countries. The reader wonders about the extent to which there also exists studies in other countries with results that differ from the ones in the present study. So please, for each result area, also mention studies in other countries which differed from the results in the current study and discuss possible reasons why the results differed (otherwise the Discussion will have a clear tendency to “confirmation bias”!)! Possibly a smartly designed table can be used to show which studies gave similar results and which studies gave different results from the present ones.

L 324: words etc. missing in this sentence!

L 354: What is meant by “significantly high” here? Can “significantly” be deleted?

Please give suggestions for future research in the Discussion section!

Please give concrete suggestions for how the results of this study can be used to improve the Ethiopian prisons!

Please provide concrete information about the contribution of this study!

6. PLOS authors have the option to publish the peer review history of their article (what does this mean?). If published, this will include your full peer review and any attached files.

Reviewer #1: **Yes: **Denis Rustand

Reviewer #2: No

---

## [Author Response · Author response to Decision Letter 0]

27 Feb 2024

Author's Response to Editor(s) and Reviewers’ Comments

Manuscript ID: PONE-D-23-21802

Manuscript title: Suicidal ideation, attempt and associated factors among prisoners in Northwest Ethiopia: A cross-sectional study.

Dear editor(s) and reviewers

 First, the authors would like to thank the editor and reviewers for your precious time, thoughtful comments, and constructive suggestions, which help to improve the quality of this manuscript. We have responded to each comment and believe that the manuscript is much improved with the changes we made as suggested by the editors and reviewers. The corresponding changes made in the revised manuscript are summarized in our response below. Herewith, please find, in bold font (Track), the point-by-point response to the reviewer’s comments and suggestions.

With kind regards! 

On behalf of the co-authors 

Setegn Fentahun, the correspondence author.

Reviewer #1: 

Title: “Suicidal ideation, attempt and associated factors among prisoners in Northwest Ethiopia: a cross-sectional study”

The paper evaluates the prevalence and associated factors of suicidal ideation and attempt among prisoners in Northwest Ethiopia in 2022 in order to inform health policies. The main statistical analysis is based on logistic regression.

This work is well-motivated and interesting for health research. As my expertise is limited to the methodology, there is not much to say as the statistical methodology is quite simple. The references needs to be double checked and some needs to be fixed. My comments and suggestions for possible modifications are below.

Comments

1. Abstract - Methods: The authors state "a simple random sampling technique was employed", this sounds too vague.

Response: Dear reviewer thank you very much for your constructive comments. We have made a revision on the revised manuscript.

2. Abstract - Methods: "Binary and multivariate analyses were used". The authors could be a bit more specific here.

Response: We thank you. The correction has been made to the abstract section.

3. Introduction: The reference for "Over 800,000 people worldwide kill themselves each year (8)" is not correct, I believe it should be reference number 9.

Response: We have changed the reference as you suggested.

4. Introduction: The reference for "Suicide rates in prison are seven times higher than in the general population (8)" does not seem to contain this information.

Response: we have checked the reference

5. Introduction: Same comment for "In the United States, jailed people are roughly four times more likely to commit suicide than the general population, and nearly half of the prisoners have had suicidal ideation or conduct at some point in their lives, and one in every five has attempted suicide (10)."

Response: We thank you very much for your suggestions. We checked the reference, and we have replaced the reference that contains the above information.

6. Introduction: "Suicide rates in prison are seven times higher than in the general population" and" The suicide rate in prison is higher than that of people of a similar age and sex who live in the community, with a [...] two-fold increase in the actual suicide rate (11)." It seems inconsistent, this should be discussed...

Response: Thank you so much. A revision has been made on the manuscript by cancelling the inconsistent sentences. 

7. Introduction: References 19 and 20 are identical to reference 15.

Response: we revised these references. 

8. Sample size determination: "Including 10% of the non-response rate the final sample size was 730+73=803".

If you want to have 730 responses with an expected 10% of non-response rate, then you should do 730/0.9=811 instead of multiplying by 1.10 as 803 - 10% = 723...

Response: Thank you very much for the recommendation of this sample size formula. We accept your comment. We will use the above formula for our future work. 

9. Discussions: "This is due to a parental history of mental illness might increase the risk of suicidal ideation in the offspring through transmission of genetic factors (8)". Check the reference. 

Response: We have checked the reference.

10. Conclusion: "the magnitude of suicidal ideation and attempt among prisoners was significantly high." I'm not sure of the meaning of "significantly high" here, maybe the authors should reformulate.

Response: Thank you very much. The correction has been made on the revised manuscript. The prevalence of suicidal ideation and attempt among prisoners was high compared to the general population. 

11. It is not clear if STATA was used for the logistic regression or just for descriptive statistics, this should be clarified.

Response: Thank you for your concern. We have made the revision on the method section (data processing and analysis) of the revised manuscript.

12. Possible interactions between factors in multivariate logistic models are not discussed and could be of interest.

Response: We thank you. It is good to conduct the interaction of independent factors in the multivariate logistic model. However, it was not our main objective. Our objectives were to determine the prevalence of suicidal ideation and suicidal attempt among prisoners and to identify factors associated with suicidal ideation and suicidal attempt among prisoners. 

Reviewer #2: 

This research investigates various variables associated with ideations and attempts at suicide among prisoners in three prisons in North-West Ethiopia and the prevalence of ideations and attempts at suicide among the same prisoners. This research is in the main well conducted but several things should be better explained and the English in the text could be improved at places in the text. The weakest part of the text is, I think, the Discussion section which should be improved.

Line 91 (= L 91): delete the words “in the study areas”? Redundant!

Response: Dear reviewer thank you very much for your constructive comments. We have deleted the words as you suggested. 

It would be good if the authors could provide more info about the prison system I Ethiopia and NW Ethiopia and how the standard and routines of these compare to prisons in other more or less developed countries. 

Moreover, I also missed a discussion in Discussion section how the standard of the prisons might have contributed to the suicide ideations and attempts. Please provide such a discussion!

Response: The details about the Ethiopian prison system and mental health services in Ethiopian prisons are added to the introduction section of the main manuscript. We also included the contributions of the standards of Ethiopian prisons to the suicide ideation and attempts in the discussion section, as you suggested.

L 107-108: Please provide info about How many prisoners there were there in total? And what it means that they were “included in the study”?

Response: We thank for the comments. The revision has been made in the clean version of revised manuscript as you recommended. Included in this study means we mentioned here the inclusion and exclusion criteria.

L 114-115: incomplete sentence!

Response: We have made the correction.

L 115-122: This needs to be explained much better!

Response: Thank you very much. We added details about study selection in the sampling procedure section.

L 175: Why not “Results”?

Response: We accept the comments.

L 177: Explain carefully what is meant by “response rate” here! For example, response rate in ration to what?

Response: We thank you. We calculated the response rate from the total sample size (803). After the data was collected, out of 803 prisoners, only 788 study participants fully responded to the questionnaires. 

L 193: How can 45.7 % be “a majority”?

Response: The correction has been made to the revised manuscript. 

L 204: Since the prisoners were asked about suicide ideation and attempts only in the “last month” it seems likely that the rates of these measures will be under-reported if they are interpreted to be valid for a whole year. I assume the questions were asked like this for memory reasons, but the drawbacks and consequences should be discussed in the Discussion and under “limitations”!

Response: Thanks. We included the recall bias in the limitation section.

Moreover, on L 243-244, given that they were asked about suicide ideation and attempts only in the “last month”, it does not seem appropriate to write about the “prevalence of suicidal ideation and attempt during their current imprisonment”.

Similarly, on L 261-262 it is not appropriate to write about “the entire period after their imprisonment”.

Response: The amendments have been made to the revised manuscript. As we mentioned in the result section, we assessed the one-month prevalence of both suicidal ideation and attempt (The prevalence of suicidal ideation and attempt in the last month was 65 (8.3%) and 27 (3.4%), 205 respectively). We also assessed suicidal ideation and attempt after their imprisonment, and finally, we discussed using this prevalence (after imprisonment). Since we assessed the overall suicidal ideation and attempt during the entire period of their current imprisonment, we mentioned the recall bias as a limitation in the discussion section.

L 214 and L 216: Explain why different significance levels are used in these contexts!

Response: It is good to include all variables in multivariate logistic regression analysis, but if we include all variables in multivariate analysis, it may be exposed to confounding (multicollinearity) and affect our final model. It may also prohibit the association of necessary variables with the outcome.

L 241: should be “Discussion”!

Response: Thank you so much. We have mad the correction.

L262: add the word “be”, or similar, in this sentence.

Response: We added it.

L 262-264: Poor grammar!

Response: We have checked the grammar, and the amendment has been done.

L 284-286: Too vague! Explain better how the difference could be due to “different instruments”! and “variation in study subjects”!

Response: Thank you very much. The revision has been made as you suggested. 

In the Discussion the results of the current study are related to previous research and then it is noted that they are similar to results from diverse studies in other countries. The reader wonders about the extent to which there also exists studies in other countries with results that differ from the ones in the present study. So please, for each result area, also mention studies in other countries which differed from the results in the current study and discuss possible reasons why the results differed (otherwise the Discussion will have a clear tendency to “confirmation bias”!)! Possibly a smartly designed table can be used to show which studies gave similar results and which studies gave different results from the present ones.

Response: Thanks. We have revised the discussion section.

L 324: words etc. missing in this sentence!

Response: We have made the revision to the sentence.

L 354: What is meant by “significantly high” here? Can “significantly” be deleted?

Response: We thank you for your concern. We accept the comment.

Please give suggestions for future research in the Discussion section!

Please give concrete suggestions for how the results of this study can be used to improve the Ethiopian prisons!

Please provide concrete information about the contribution of this study!

Response: Thanks a lot. We have included the above points in the conclusion and recommendation section of the manuscript.

---

## [Decision Letter · Decision Letter 1]

18 Mar 2024

Suicidal ideation, attempt and associated factors among prisoners in Northwest Ethiopia: A cross-sectional study

PONE-D-23-21802R1

Dear Dr. Fentahun,

We’re pleased to inform you that your manuscript has been judged scientifically suitable for publication and will be formally accepted for publication once it meets all outstanding technical requirements.

Kind regards,

Vincenzo De Luca

Academic Editor

PLOS ONE

Additional Editor Comments (optional):

Reviewers' comments:

Reviewer's Responses to Questions

**Comments to the Author**

1. If the authors have adequately addressed your comments raised in a previous round of review and you feel that this manuscript is now acceptable for publication, you may indicate that here to bypass the “Comments to the Author” section, enter your conflict of interest statement in the “Confidential to Editor” section, and submit your "Accept" recommendation.

Reviewer #1: All comments have been addressed

Reviewer #2: (No Response)

2. Is the manuscript technically sound, and do the data support the conclusions?

Reviewer #1: (No Response)

Reviewer #2: Yes

3. Has the statistical analysis been performed appropriately and rigorously? 

Reviewer #1: (No Response)

Reviewer #2: I Don't Know

4. Have the authors made all data underlying the findings in their manuscript fully available?

Reviewer #1: (No Response)

Reviewer #2: Yes

5. Is the manuscript presented in an intelligible fashion and written in standard English?

Reviewer #1: (No Response)

Reviewer #2: Yes

6. Review Comments to the Author

Reviewer #1: (No Response)

Reviewer #2: The researchers have done a good job when revising their ms. It is now ready for publication when the authors have:

1.Explain the formula on line 138 : All the symbols should be explained, and it should be explained how the symbols “translate” n to the formula with the numbers on the same line!

2.Line 139: explain what is meant by “non-response rate” (I may have missed it!)

3.Correct the grammar for line 317: for ex to “… was higher than other studies conducted in different countries.”

4.Correct the grammar for line 378: for ex to: “… found to be a factor associated with more suicide attempts, maybe because of …”

7. PLOS authors have the option to publish the peer review history of their article (what does this mean?). If published, this will include your full peer review and any attached files.

Reviewer #1: No

Reviewer #2: No

---

## [Editor Report · Acceptance letter]

29 Mar 2024

PONE-D-23-21802R1 

PLOS ONE

Dear Dr. Fentahun, 

I'm pleased to inform you that your manuscript has been deemed suitable for publication in PLOS ONE. Congratulations! Your manuscript is now being handed over to our production team.

Kind regards, 

on behalf of

Dr. Vincenzo De Luca 

Academic Editor

PLOS ONE